# Epigenetic Silencing of Tumor Suppressor lncRNA *NKILA*: Implication on NF-κB Signaling in Non-Hodgkin’s Lymphoma

**DOI:** 10.3390/genes13010128

**Published:** 2022-01-11

**Authors:** Min-Yue Zhang, George Calin, Ming-Dan Deng, Rex K. H. Au-Yeung, Lu-Qian Wang, Chor-Sang Chim

**Affiliations:** 1Department of Medicine, Queen Mary Hospital, The University of Hong Kong, Hong Kong, China; zhangminyue@connect.hku.hk (M.-Y.Z.); u3003939@connect.hku.hk (M.-D.D.); wanglucy@hku.hk (L.-Q.W.); 2Division of Hematology, School of Medicine, Renji Hospital, Shanghai Jiaotong University, Shanghai 200127, China; 3Department of Experimental Therapeutics, The University of Texas MD Anderson Cancer Center, Houston, TX 77030, USA; gcalin@mdanderson.org; 4Department of Pathology, Queen Mary Hospital, The University of Hong Kong, Hong Kong, China; rex.auyeung@hku.hk

**Keywords:** DNA methylation, *NKILA*, non-Hodgkin’s lymphoma, tumor suppressor, NF-κB signaling

## Abstract

The long non-coding RNA (lncRNA) *NKILA,* localized to 20q13.31, is a negative regulator of NF-κB signaling implicated in carcinogenesis. As a CpG island is embedded in the promoter region of *NKILA*, it is hypothesized as a tumor suppressor lncRNA silenced by promoter DNA methylation in non-Hodgkin’s lymphoma (NHL). By pyrosequencing-verified methylation-specific PCR, *NKILA* methylation was detected in 1/10 (10%) NHL cell lines, but not in normal peripheral blood buffy coats or tonsils. *NKILA* methylation correlated with the repression of *NKILA* in cell lines. Hypomethylation treatment with 5-Aza-2′-deoxycytidine resulted in promoter demethylation and the re-expression of *NKILA*. In 102 NHL primary samples, *NKILA* was methylated in 29 (51.79%) diffuse large B-cell lymphoma (DLBCL) and 4 (20%) peripheral T-cell lymphoma cases, but unmethylated in all 26 mantle cell lymphoma cases. Mechanistically, the knockdown of *NKILA* resulted in promoting IkBα phosphorylation, associated with nucleus translocation of total p65 and phosphorylated p65 in SU-DHL-1 cells, hence constitutive NF-κB activation. Functionally, the knockdown of *NKILA* in SU-DHL-1 cells led to decreased cell death and increased cellular proliferation. Collectively, *NKILA* was a tumor suppressor lncRNA frequently hypermethylated in DLBCL. Promoter DNA methylation-mediated *NKILA* silencing resulted in increased cellular proliferation and decreased cell death via the repression of NF-κB signaling in NHL.

## 1. Introduction

Non-Hodgkin’s lymphoma (NHL) encompasses a heterogeneous group of diseases, including B-cell, T-cell and natural killer (NK)-cell lymphoma based on the origin and lineage of tumor cells [1]. B-cell lymphoma comprises more than 70% of all lymphomas, while T-cell lymphoma accounts for 10–15% of all lymphomas [2]. Diffuse large B-cell lymphoma (DLBCL) and follicular lymphoma are the two most common subtypes of NHL [3], whereas NK-cell lymphoma is an aggressive subtype rare in Western countries [4]. The incidence of NHL was 6.7/100,000 among males and 4.7/100,000 among females worldwide, which ranked as the 8th and 10th of all types of cancers, respectively [5]. The clinical features of NHL patients include painless lymphadenopathy, organomegaly and the presence or absence of B symptoms (night sweats, weight loss > 10% and fever with temperature > 38 °C) [6].

DNA methylation refers to the addition of methyl group (–CH_3_) to C5 position of cytosine in a CpG dinucleotide via DNA methyltransferases [7]. Cancer cells are characterized by two major alterations of DNA methylation: global DNA hypomethylation but the gene-specific DNA hypermethylation of promoter-associated CpG island [8]. Moreover, promoter DNA hypermethylation-mediated reversible silencing of tumor suppressor protein-coding genes, including *PTPL1* [9] and *SHP1* [10,11], and tumor suppressive microRNAs (miRs), including *miR-155-3p* [12] and *miR-146a* [13], has been implicated in lymphomagenesis.

Long non-coding RNAs (lncRNAs) are broadly defined as a class of non-coding RNAs measuring >200 nucleotides [14,15]. lncRNAs play essential roles in regulating multiple biological processes, including cellular metabolism, organogenesis and carcinogenesis [16,17,18]. LncRNA, *NKILA* (NF-KappaB Interacting LncRNA), localized to 20q13.31, was firstly found to be downregulated in breast cancer. Overexpression of *NKILA* resulted in the inhibition of metastasis and the increase in apoptosis by repression of NF-κB signaling activity in breast cancer cells, indicating the tumor suppressor property of *NKILA* [19]. Moreover, the NF-κB signaling pathway is constitutively activated and, hence, implicated in the pathogenesis of NHL. However, the function of *NKILA* in lymphoma remains unknown.

As a CpG island is present at the promoter region of *NKILA*, we postulated that *NKILA* is a tumor suppressor lncRNA reversibly silenced by promoter DNA methylation in NHL. Herein, the methylation of *NKILA* will be studied in NHL, and the role of *NKILA* in lymphomagenesis will also be investigated.

## 2. Materials and Methods

### 2.1. Patient Samples

A total of 102 formalin fixed, paraffin-embedded (FFPE) or fresh frozen diagnostic lymph node biopsy tissues, including 56 DLBCL, and 26 mantle cell lymphoma (MCL) and 20 peripheral T-cell lymphoma (PTCL) cases, were acquired from 5 hospitals in Hong Kong, China (Queen Mary Hospital, Kwong Wah Hospital, Princess Margaret Hospital, United Christian Hospital and Pamela Youde Nethersole Eastern Hospital). The diagnosis of lymphoma was based on the WHO (World Health Organization, Geneva, Switzerland) classification [20]. Eleven FFPE tonsil tissues were also obtained from healthy individuals undergoing tonsillectomy. Our study was approved by the Institutional Review Board of Queen Mary Hospital.

### 2.2. Cell Culture

Five MCL cell lines (SP53, REC-1, GRANTA-519, MINO and JEKO-1), two DLBCL cell lines (SU-DHL-6 and SU-DHL-16), two ALK (+) anaplastic large cell lymphoma (ALCL) cell lines (KARPAS-299 and SU-DHL-1) and one T-cell lymphoblastic lymphoma cell line (SUP-T1) were used in this study. SP53 and REC-1 were kindly provided by Professor Raymond Lai (Department of Laboratory Medicine and Pathology, University of Alberta and Cross Cancer Institute). Other cell lines were purchased from Deutsche Sammlung von Mikroogranismen und Zellkulturen (DSMZ) (Braunschweig, Germany). Cell lines were maintained in RPMI-1640 (DMEM for GRANTA-519) supplemented with 10–15% fetal bovine serum, 50 U/mL of penicillin and 50 ug/mL streptomycin in a humidified atmosphere of 5% CO_2_ at 37 °C.

### 2.3. DNA Demethylation Treatment

SU-DHL-6 cells (1 × 10^6^ cells/mL) were seeded in 25 cm^2^ flasks and cultured with 0.5–1.5 μM of 5-aza-2′-deoxycytidine (5-azadC) (Sigma-Aldrich) for 7 days. The 5-azadC was replaced every 24 h. Afterwards, the cells were harvested for DNA and RNA extraction.

### 2.4. DNA Extraction

DNA from NHL cell lines and healthy peripheral blood was extracted with a DNA Blood Mini kit (Qiagen). DNA extraction from frozen patient biopsies was conducted with an automated DNA extraction system (DNA Tissue Kit from Qiagen). DNA extraction from FFPE tissues were performed by using QIAamp DNA FFPE Tissue Kit (Qiagen).

### 2.5. Methylation-Specific Polymerase Chain Reaction (MSP)

Sodium bisulfite conversion was conducted with an EpiTect Bisulfite Kit (Qiagen). Afterwards, MSP was performed in bisulfite-treated DNA with two sets of primers, which were specific to unmethylated (U-MSP) or methylated (M-MSP) DNA sequences. MSP primers were designed at the CpG island upstream to the *NKILA* gene by the online tool Methprimer (http://www.urogene.org/methprimer/ (accessed on 31 May 2019). Details of the primer sequence and PCR condition for MSP are listed in Table 1. For each MSP reaction, the enzymatically methylated control DNA (CpGenome Universal Methylated DNA; Chemicon/Millipore, Billerica, MA, USA) was used as the positive control for M-MSP and the negative control for U-MSP.

### 2.6. Quantitative Reverse Transcription Polymerase Chain Reaction (qRT-PCR)

Total RNA was isolated with a Direct-zol™ RNA MiniPrep kit (Zymo Research), followed by reverse transcription with SuperScript^®^ III (Invitrogen, Carlsbad, CA, USA). The qRT-PCR was performed with a SYBR^®^ Select Master Mix (ABI, Bedford, MA, USA), and the human glyceraldehyde 3-phosphate dehydrogenase (*GAPDH*) was used as the endogenous control. The relative quantity of *NKILA* expression was calculated by the method of 2^−ΔΔCt^ and normalized against the endogenous control. The primer sequences for *NKILA* and *GAPDH* are listed in Table 1 [19,21].

### 2.7. Quantitative Bisulfite Pyrosequencing

The promoter region of *NKILA* overlapped with the amplicon of MSP was amplified in the bisulfite converted DNA with methylation-unbiased primers. Primer sequences and condition for PCR were listed as follows: forward primer, 5′-GTT GGG GAG AGG GTA TAG-3′; reverse primer, 5′-Biotin-CTC CTC CTC CTC ATT CAA ATC-3′ and condition, 56 °C/45 x/1.5 mM MgCl_2_. A Qiagen PyroMark PCR Kit was employed to perform PCR amplification. Afterwards, a stretch of DNA containing 10 consecutive CpG dinucleotides were pyrosequenced with a sequencing primer, 5′-GTT AGG GGA GGG GGT G-3′, on a PSQ 96MA system and analyzed using PyroQ-CpG 1.0.9. software.

### 2.8. Knockdown of NKILA

RNA interference by small interfering RNA (siRNA) was used to knockdown the expression of *NKILA*. Briefly, SU-DHL-1 cells were seeded at a density of 0.5 × 10^6^/mL in a six-well plate and transfected with *NKILA* siRNA (n541256, Ambion, Austin, TX, USA) or Silencer Negative Control (Ambion) at a final concentration of 150 nM with an RNAiMAX transfection reagent (Invitrogen, Carlsbad, CA, USA). The transfected cells were cultured for 24 h or 48 h.

### 2.9. Cell Proliferation and Cell Death

After 24 h and 48 h transfection, viable cells were analyzed by trypan blue exclusion and the cell number was measured by the Countess II (Invitrogen). Cell proliferation was calculated by normalization with the negative control at 48 h post-transfection. Cell death was analyzed by trypan blue exclusion at 24 h post-transfection. All experiments were repeated at least three times.

### 2.10. Western Blot

SU-DHL-1 cells transfected with *NKILA* siRNA and Silencer Negative Control were treated with 200 ng/mL TNFα for 1 hour before being lysed in RIPA buffer (Cell signaling). Totally, 15 µg protein of each sample was loaded and separated in Mini-PROTEAN TGXTM Precast gel (Bio-rad, Hercules, CA, USA), followed by transference onto a 0.45 µm PVDF membrane (Amersham, Chiltern, Buckinghamshire, UK). The membrane was blocked in a super blocking buffer and incubated in primary antibodies, including p65 (1:1000, cell signaling); p-p65 ser536 (1:1000, cell signaling); IkBα (1:1000, cell signaling); p-IkBα ser32 (1:1000, cell signaling); β-tubulin (1:1000, cell signaling) and nucleolin (1:1000, cell signaling) at 4 °C overnight with a gentle rotation. The membrane was then washed and incubated with HRP-linked secondary antibodies, anti-mouse (1:3000, cell signaling) and anti-rabbit (1:3000, cell signaling) for one hour, followed by TBST washing. The ECL HARP substrate was used on the membrane before being developed in X-ray films.

Nuclear and cytosol fractionations of SU-DHL-1 cells were prepared by the Nuclear/Cytosol Fractionation Kit (BioVision, Milpitas, CA, USA), according to the manufacturers’ instructions. Western blot was then performed in these samples.

### 2.11. Statistical Analysis

The difference of cell proliferation and cell death between SU-DHL-1 cells transfected with *NKILA* siRNA and Silencer Negative Control were compared by Student’s *t*-test. The difference of the *NKILA* methylation frequency in different subtypes of NHL primary samples was analyzed by the χ^2^ test. All *p*-values were 2-sided; *p* < 0.05 was considered as the significant difference.

## 3. Results

### 3.1. NKILA Was Methylated in a Tumor-Specific Manner in NHL Cells

*NKILA* was reported to repress the NF-κB signaling pathway [19,22,23], which is constitutively activated and implicated in lymphomagenesis. There is a CpG island at the promoter region of *NKILA*. Hence, the methylation status of *NKILA* was investigated by MSP in the bisulfite-converted DNA of normal healthy controls, including 10 peripheral blood buffy coats and 11 normal tonsil tissues, in addition to 10 NHL cell lines. Direct sequencing of M-MSP products from methylated positive control DNA demonstrated that all unmethylated cytosines were converted into thymidines after PCR, whereas all methylated cytosines remained unchanged, indicating the complete bisulfite conversion and specificity of MSP (Figure 1A). By MSP, the methylation of *NKILA* was absent in all of normal peripheral blood buffy coats and normal tonsil tissues (Figure 1B). Amongst NHL cell lines, *NKILA* was completely methylated (MM) in SU-DHL-6 and completely unmethylated (UU) in GRANTA-519, JEKO-1, MINO, REC-1, SP-53, KARPAS-299 and SU-DHL-1. However, neither U-MSP nor M-MSP signals were observed in SUP-T1 cells (Figure 1C). Furthermore, the methylation status of *NKILA* in NHL cell lines detected by MSP was verified by quantitative bisulfite pyrosequencing. SU-DHL-6 cells with a complete methylation of *NKILA* had a mean methylation percentage of 69.4%. In contrast, NHL cell lines with complete unmethylation of *NKILA* had a mean methylation percentage ranging from 5.0% to 6.5%, which confirmed the methylation status detected by MSP (Figure 1D). These results indicated that *NKILA* was methylated in a tumor-specific manner in NHL cells.

### 3.2. The Expression of NKILA Was Inversely Correlated with Promoter DNA Methylation

To explore the relationship between promoter DNA methylation and the expression of *NKILA*, semi-quantitative RT-PCR of *NKILA* was performed in NHL cell lines. As demonstrated by the DNA gel, no expression of *NKILA* was detected in SU-DHL-6 cells that was completely methylated for *NKILA*. Conversely, the expression of *NKILA* was observed in other cell lines completely unmethylated for *NKILA* (Figure 2A).

Furthermore, to study whether promoter DNA methylation was associated with reversible silencing of *NKILA*, SU-DHL-6 cells, which were completely methylated for *NKILA*, were treated with a demethylating agent, 5-AzadC, for 7 days. Upon treatment with 5-AzadC, the promoter of *NKILA* was demethylated, as illustrated by the emergence of U-MSP signal (Figure 2B), with the re-expression of *NKILA* (Figure 2C). Hence, these data suggested that the reversible silencing of *NKILA* was mediated by promoter DNA methylation in NHL cells.

### 3.3. NKILA Was Differentially Methylated in NHL Primary Samples

To investigate the methylation of *NKILA* in NHL primary samples, MSP was performed with bisulfite-converted DNA in primary samples, including 26 MCL, 56 DLBCL and 20 PTCL cases. The MSP results showed that no methylation of *NKILA* was detected in primary MCL samples (Figure 3A). However, *NKILA* was found to be methylated in 29 (51.79%) DLBCL and 4 (20%) PTCL cases (Figure 3B,C), hence preferentially methylated in DLBCL than MCL (*p* < 0.0001) and PTCL (*p* = 0.007).

### 3.4. NKILA Inhibited IkBα Phosphorylation and NF-κB Activation

*NKILA* has been reported to suppress NF-κB signaling pathways by blocking IkBα phosphorylation in breast cancer, non-small cell lung cancer and nasopharyngeal carcinoma [19,22,23]. To elucidate *NKILA* function in lymphoma, *NKILA*-targeted siRNA was used to knockdown *NKILA* and non-targeting siRNA was used as a control in SU-DHL-1 cells. The qRT-PCR result confirmed that *NKILA* was knocked down at 24 h and 48 h post-transfection (Figure 4A). To further evaluate *NKILA* function in NF-κB signaling pathways, we examined IkBα phosphorylation and p65 nucleus translocation after *NKILA* knockdown, with an augmentation of the NF-κB signaling by TNFα [19]. The result showed that *NKILA* knockdown led to an increase in IkBα phosphorylation in SU-DHL-1 cells at both 24 h and 48 h post-transfection (Figure 4B), which was associated with enhanced p65 being translocated into the nucleus, compared with the control. Moreover, the enhanced nuclear translocation of phosphorylated p65 ser536 was observed after *NKILA* knockdown (Figure 4C). These results collectively indicated that *NKILA* negatively regulated NF-κB signaling pathways, by inhibiting IkBα phosphorylation and reducing nuclear translocation of total and phosphorylated p65.

### 3.5. NKILA Was a Tumor Suppressor lncRNA in SU-DHL-1 Cells

As *NKILA* was a negative regulator in NF-κB signaling pathways, its tumor suppressor function in lymphoma was further explored by examining cell proliferation and cell death in SU-DHL-1 cells that were completely unmethylated for *NKILA*. The knockdown of *NKILA* led to a significantly increased cell proliferation rate compared with the control (Figure 5A). Furthermore, the knockdown of *NKILA* resulted in reduced cell death in SU-DHL-1 cells (Figure 5B). These results supportively indicated *NKILA* acted as a tumor suppressor lncRNA in SU-DHL-1 cells.

## 4. Discussion

Several observations were made in this study. Firstly, we showed that *NKILA* was methylated in NHL cell lines and NHL primary samples, but unmethylated in normal controls, hence methylated in a tumor-specific manner in NHL. This was consistent with a tumor-specific pattern of the methylation of tumor suppressor protein-coding genes, such as *p16* and *p15* [24], and non-coding tumor suppressor microRNAs, such as *miR-342-3p* [25] and *miR-1250-5p* [26], in NHL. However, this contrasted with the tissue- but not tumor-specific pattern of methylation, such as *miR-373* [27] and *miR-127* [28], which were shown to be methylated in both normal counterparts and tumor cells, hence likely unimportant in carcinogenesis.

Secondly, in primary NHL samples, *NKILA* was frequently methylated in DLBCL, but not in MCL or PTCL samples. Indeed, NHL is highly heterogenous with different genetic and epigenetic features [1,20]. For instance, a microarray-based DNA methylation study in 367 hematological neoplasms demonstrated that promoter DNA hypermethylation was more frequent in precursor B and T lymphoid neoplasias and mature B-cell lymphomas of a germinal center origin (such as DLBCL, FL and Burkitt’s lymphoma), than in mature T-cell lymphomas, such as PTCL [29]. Therefore, differential methylation of *NKILA* can be accounted for the difference in cell origin and their inherent pathogenetic mechanisms.

Thirdly, to our knowledge, this is the first study that reported that promoter DNA methylation mediated the reversible silencing of *NKILA* in NHL, which is evidenced by an inverse correlation between the *NKILA* methylation and its expression in NHL cell lines, and the re-expression of *NKILA* upon demethylation treatment in cell lines with a complete methylation of *NKILA*. Apart from promoter DNA methylation, *NKILA* have been reported to be regulated by other mechanisms. For instance, Huang et al. [30] showed that, in cytotoxic T lymphocytes, *NKILA* transcription induced by antigen stimulation was mediated by an increase in the acetylation of histones (H4ac, H3K27ac and H3K9ac) at the promoter region of *NKILA,* suggesting the expression of *NKILA* regulated by histone modification. In addition, in breast cancer cells, some oncogenic microRNAs, such as *miR-103* or *miR-107*, could directly target and downregulate the expression of *NKILA*, which was confirmed by the luciferase reporter assay [19].

Fourthly, *NKILA* was first shown to suppress NF-κB signaling in breast cancer. It can directly bind to the NF-κB/IκB complex, and potentially mask the phosphorylation site of IκB, thereby suppressing IKK-induced IκB phosphorylation and, hence, NF-κB activity [19]. As the constitutive activation of the NF-κB signaling pathway is the hallmark of many lymphoid malignancies, including Hodgkin’s lymphoma and DLBCL [31], and hence implicated in lymphomagenesis, it is essential to comprehend the precise function of *NKILA* and its interaction with NF-κB in lymphomas. To the best of our knowledge, this is the first study showing the role of *NKILA* in lymphomas. Previous studies reported that phosphorylation p65 at ser536 led to the enhanced transcriptional activity of NF-κB [32,33,34]. We observed that the knockdown of *NKILA* led to the upregulation of NF-κB signaling pathways by promoting IκBα phosphorylation, and the consequent nuclear translocation of total p65 and phosphorylated p65 in lymphoma cells. Furthermore, as a transcription factor, NF-κB regulates the expression of multiple downstream effectors, enhancing cell proliferation, inhibiting cell apoptosis or promoting cell migration and invasion [35]. Herein, in NHL cells, the downregulation of *NKILA* resulted in an increase in cellular proliferation and decrease in cell death, consistent with the tumor suppressor role of *NKILA* in multiple tumors, including melanoma, lung cancer, rectal cancer, laryngeal cancer and breast cancer [19,22,36,37,38,39,40]. In addition, *NKILA* has also been shown to inhibit tumor invasion and migration in epithelial cancers, such as breast cancer, hepatocellular carcinoma and tongue squamous cell carcinoma [19,39,40]. Collectively, these results suggested that the *NKILA* can inhibit cellular proliferation and induce cell death by suppressing the NF-κB signaling pathway in NHL cells.

## 5. Conclusions

Taken together, the epigenetic silencing of lncRNA *NKILA* was mediated by promoter DNA methylation in a tumor-specific manner in NHL. Frequent hypermethylation of *NKILA* was preferentially detected in DLBCL patients. *NKILA* exerted its tumor suppressive property by the inhibition of cellular proliferation and increase of cell death, in association with suppression of the NF-κB signaling pathway via inducing IκBα phosphorylation, and the consequent nucleus translocation of total p65 and phosphorylated p65 in NHL cells.

## Figures and Tables

**Figure 1 genes-13-00128-f001:**
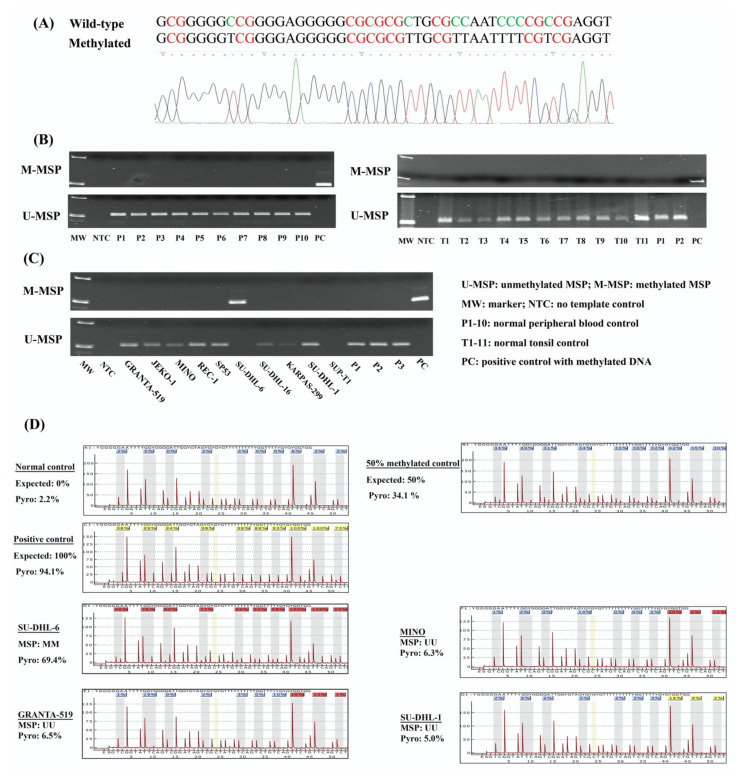
Methylation of *NKILA* in normal controls and NHL cell lines. (**A**) Direct sequencing of M-MSP products from a positive control with methylated DNA. (**B**) M- and U-MSP demonstrate that *NKILA* is unmethylated in normal peripheral blood buffy coats (P1–P10) and normal tonsil tissues (T1-T11). (**C**) M- and U-MSP show that *NKILA* is completely methylated in SU-DHL-6 cell lines and completely unmethylated in GRANTA-519, JEKO-1, MINO, REC-1, SP-53, KARPAS-299 and SU-DHL-1 cell lines. Both M-MSP and U-MSP are absent in SUP-T1 cell lines. (**D**) Quantitative bisulfite pyrosequencing analysis show the mean methylation percentage of 10 neighboring CpG dinucleotides overlapping with MSP amplicon in 0%, 50% and 100% methylation control, and NHL cell lines.

**Figure 2 genes-13-00128-f002:**
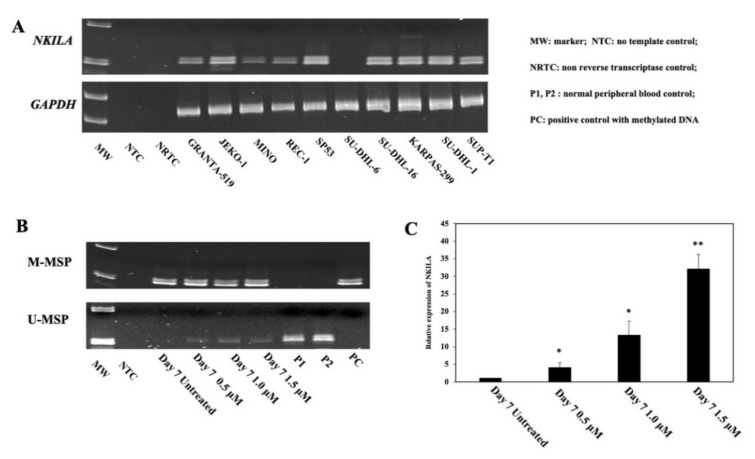
Correlation of promoter DNA methylation of *NKILA* and its expression in NHL cell lines. (**A**) *NKILA* methylation measured by MSP is associated with a lower expression of *NKILA* in NHL cell lines by semi-quantitative RT-PCR. (**B**,**C**) Treatment of SU-DHL-6 cells completely methylated for *NKILA* with 5-AzadC for 7 days, leads to *NKILA* promoter demethylation detected by MSP (**B**) and associated with the re-expression of *NKILA* by qRT-PCR (**C**). Columns represent mean +/− 1SD from three qRT-PCR experiments in triplicate. *: Compared with untreated cells, *p*-value < 0.05, **: Compared with untreated cells, *p*-value < 0.01.

**Figure 3 genes-13-00128-f003:**
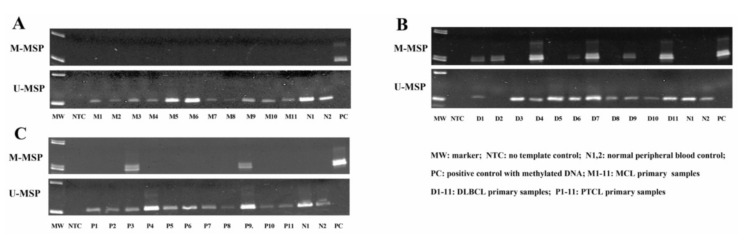
Methylation status of *NKILA* is demonstrated in different types of NHL primary samples by MSP. Representative M- and U-MSP shows the methylation of *NKILA* in MCL (**A**), DLBCL (**B**) and PTCL (**C**).

**Figure 4 genes-13-00128-f004:**
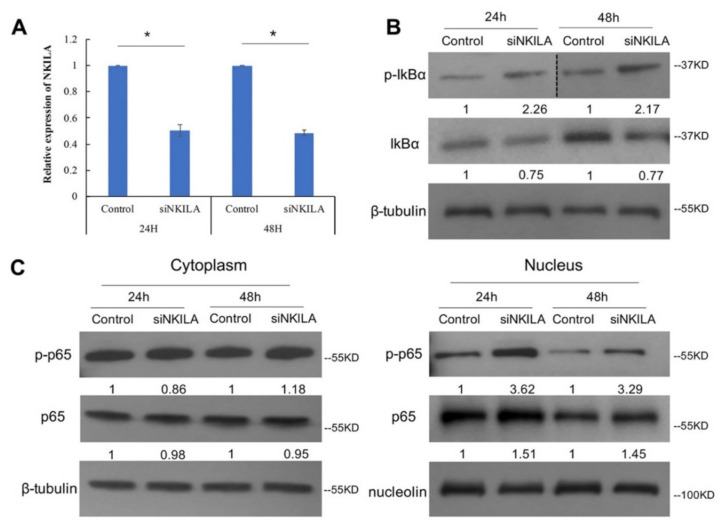
*NKILA* knockdown leads to more IκBα phosphorylation and more p65 translocated into the nucleus. (**A**) qRT-PCR was performed to examine *NKILA* knockdown efficiency in SU-DHL-1 at 24 h and 48 h post-transfection. Columns represent mean +/− 1SD from three independent experiments. *: *p*-value < 0.05. (**B**) Western blot shows the total IκBα and phosphorylated IκBα ser32 expression level. β-tubulin was used as the loading control. Quantification densitometry of each band is normalized with the loading control. p-IκBα: phosphorylated IκBα ser32. (**C**) Western blot shows the total p65 and phosphorylated p65 ser536 level in the cytoplasm and nucleus. β-tubulin and nucleolin serves as the loading control for the cytoplasm and nucleus, respectively. p-p65: phosphorylated p65 ser536; si*NKILA*: *NKILA*-targeted siRNA. 24 h: 24 h post-transfection; 48 h: 48-h post-transfection.

**Figure 5 genes-13-00128-f005:**
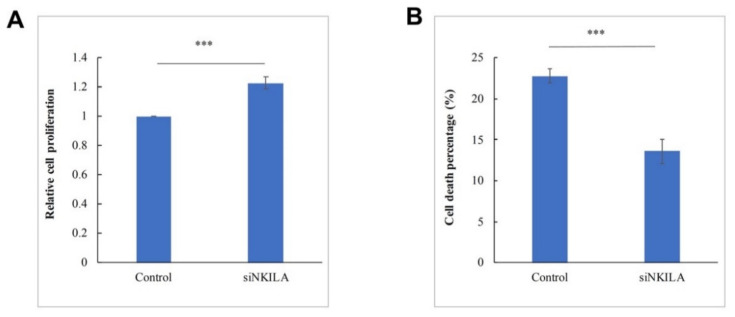
*NKILA* suppresses cell growth and promotes cell death in SU-DHL-1. (**A**) Cellular proliferation upon knockdown of *NKILA* is studied using the trypan blue assay at 48 h post-transfection. Columns represent mean +/− 1SD from five independent experiments. (**B**) Cell death upon knockdown of *NKILA* is studied using the trypan blue exclusion assay at 24 h post-transfection. Columns represent mean +/− 1SD from three independent experiments. si*NKILA*: *NKILA*-targeted siRNA. ***: *p*-value < 0.001.

**Table 1 genes-13-00128-t001:** Primer sequences and PCR reaction conditions for *NKILA*.

	Forward Primer (5′ to 3′)	Reverse Primer (5′ to 3′)	Tm/Cycles/ _2_	Reference
**Methylation-Specific PCR (MSP)**				
M-MSP	TAG GTA GAC GGT TTG ACG TTA GC	GAA AAA ACC TCG ACG AAA ATT AAC G	57 °C/35 x/2 mM	
U-MSP	GGT AGG TAG ATG GTT TGA TGT TAG T	ACA AAA AAA CCT CAA CAA AAA TTA ACA	55 °C/37 x/1.5 mM	
**Semi-Quantitative RT-PCR/Quantitative Real-Time RT-PCR**				
*NKILA*	AAC CAA ACC TAC CCA CAA CG	ACC ACT AAG TCA ATC CCA GGT G	55 °C/40 x/2 mM(Semi-Quantitative RT-PCR)	[19]
*GAPDH*	ACC ACA GTC CAT GCC ATC ACT	TCC ACC ACC CTG TTG CTG TA	60 °C/24 x/1.5 mM(Semi-Quantitative RT-PCR)	[21]

Key: M-MSP, MSP for methylated alleles; U-MSP, MSP for unmethylated alleles and Tm, annealing temperature.

## Data Availability

All data generated or analyzed during this study are included in this published article.

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
