# Peer review of "Epigenetic Silencing of Tumor Suppressor lncRNA *NKILA*: Implication on NF-κB Signaling in Non-Hodgkin’s Lymphoma"

_genes, 2022, doi:10.3390/genes13010128_

Round 1
Reviewer 1 Report
The manuscript presented by Zhang et al, entitled “Epigenetic silencing of tumor suppressor lncRNA NKILA: implication on NF-κB signaling in non-Hodgkin’s lymphoma” describes the epigenetic studies of tumor suppressor lncRNA NKILA in in non-Hodgkin’s lymphoma samples and cell lines.
The authors present data, demonstrating that promoter methylation of NKILA correlates with repression of NKILA in cell lines. Furhter, hypomethylation treatment with 5-Aza-2'-deoxycytidine resulted in promoter demethylation and re-expression of NKILA. Knockdown of NKILA resulted in promoting IkBα phosphorylation, associated with nucleus translocation of total p65 and phosphorylated p65 in SU-DHL-1 cells, hence constitutive NF-κB activation. Functionally, knock-down of NKILA in SU-DHL-1 cells decreases cell death and increases cellular proliferation. Their data suggest that NKILA was a tumour suppressor lncRNA frequently hypermethylated in DLBCL.
Some remarks that can improve the manuscript are enlisted below:
Line 15. NKILA repeats twice.
Line 17. The part “was detected in one (10%) NHL cell line SU-DHL-6” isn’t clear.
Line 20. “NKILA methylation was observed none of mantle cell lymphoma cases” needs rewriting.
Line 52. “Long non-coding RNA (lncRNA) is broadly” isn’t correct as you are not talking just for single lncRNA. Instead of that the plural form should be used.
Line 124. The sequence for NKILA siRNAs knock down is missing. Instead it should be included in M&M section.
Line 154. “The samples were then performed Western blot.” needs rewriting.
The titles in the result subsections should be more informative, giving a preliminary idea what has been observed.
Line 207. lncRNA HOTTIP appeared in the fig. 2 legend. However, it is the only place mentioned in the manuscript.
The discussion part is rather insufficient. The majority of the presented results are not discussed.
The conclusion part needs extension as well.
The primer pairs from Table 1 for semi-quantitative RT-PCR amplifies a fragment of 108 bp in ENST00000614771.2 but not in ENST00000613231.1. Why you selectively choose ENST00000614771.2 but not the shorter transcript?
Reviewer 2 Report
The authors investigated the function of long non-coding RNA (lncRNA) NKILA, and demonstrated that NKILA is involved in NHL. The experiments were overall well done. Some of the figures and figure legends need to be rewritten.
(1) Fig 2B is not qRT-PCR, but semi-quantitative RT-PCR.
(2) Fig 2C needs p-values.
(3) The figure legends for Fig3 must be more in detail. What experiments were done?
(4) Fig 4A needs p-values.
Round 2
Reviewer 1 Report
The authors answered to all remarks. The paper is suitable for publication in its current form.